# An Upper Bound Energy Formulation of Free-Chip Machining with Flat Chips and an Alternative Method of Determination of Cutting Forces without Using the Merchant's Circle Diagram

Hillol Joardar [1,*], Nitai Sundar Das [1], Barun Haldar [2,*], Kalipada Maity [3], Naser Abdulrahman Alsaleh [2] and Sabbah Ataya [2]

1. Department of Mechanical Engineering, C. V. Raman Global University, Bhubaneswar 752054, Odisha, India
2. Mechanical and Industrial Engineering Department, College of Engineering, Imam Mohammad Ibn Saud Islamic University (IMSIU), Riyadh 11432, Saudi Arabia; naalsaleh@imamu.edu.sa (N.A.A.)
3. Department of Mechanical Engineering, NIT Rourkela, Rourkela 769008, Odisha, India
* Correspondence: joardar.2011@gmail.com (H.J.); bhaldar@imamu.edu.sa (B.H.)

**Abstract:** An upper bound analysis of free-chip machining has been carried out, where the tool cutting and friction forces were determined from the deformation energy dissipated during the chip separation process. The method employed was based on the classical upper bound theorem, as formulated by Prager and Hodge, and Drucker, Prager, and Greenberg, and its modification by Collins, to deal with the metal forming processes involving coulomb friction. A straight shear plane and coulomb friction at the chip/tool interface were assumed and the energy required for cutting was calculated from a strain rate/velocity field that was constructed using the method proposed by Collins. Cutting forces, thrust forces, tool/chip contact lengths, and chip thickness ratios were determined for different tool rake angles and friction conditions. The theoretical results were also compared with some experimental results that are available in the published literature. The comparison between the two was not found to be satisfactory. This may be due to the non-unique nature of the machining process, as stated by Hill and demonstrated by other authors. The results calculated from the present method of "energy balance" were also found to be in agreement with those obtained by Merchant using the principle of "force balance".

**Keywords:** upper bound; free-chip machining; flat chips; deformation energy; tool forces



## 1. Introduction

The forces experienced by a cutting tool during a machining operation are considered important machining parameters, since they control the input power, affect the distribution of temperature in the tool and the work piece, and have considerable influence on the finish of the machined surface [1]. Hence, theoretical and experimental determinations of these forces have received considerable attention from scientists and engineers in the past six decades. The exact determination of these forces is rather difficult due to the fact that these are influenced by a number of input variables, such as the strength and hardness of the work material, the tool geometry, and the cutting conditions [1]. Hence, predictive models for cutting forces are usually developed based on simplifying assumptions. Thus, for theoretical analysis, the cutting is considered to be orthogonal. This implies that the chips produced are rectangular in shape and that the chips flow during cutting in a direction normal to the principal cutting edge. Also, the material is taken to be rigid plastic, and the strain hardening and strain rate effects are neglected.

Theoretical analysis of the metal machining process has generally been carried out using the three following methods: the slip-line field technique, the finite element method, and the upper bound analysis. Slip-line field solutions for orthogonal cutting with chip streaming has been suggested by Lee and Shaffer [2] and Kudo [3]. These authors assumed

the Coulomb friction condition at the chip/tool contact faces and provided solutions for both continuous chips and chips with a built-up-edge. Similar solutions for cutting with curled chips have been presented by Kudo [3], Dewhurst [4], Maity and Das [5], and Das and Dundur [6]. These analyses were carried out assuming shear friction (τ = mk), Coulomb friction (τ = μp) and adhesion friction, respectively. The analytically calculated cutting force and shear plane angle values were also validated with experimental data available in the literature. Solutions for cutting with controlled-contact cutting tools, for tools with secondary rake angles, and those with rounded cutting edges have been provided by Fang [7–9], while those for cutting with a tool with crater wear was outlined by Long and Huang [10]. Predictive models with experimental validation for machining using restricted contact tools and grooved tools were also presented by Fang and Jawahir [11,12]. Discussion on the characteristics and the mechanism of formation of segmental chips may be found in the studies published by the authors of [1,13–17]. Solutions accounting for elastic contact has been presented by Das and Dundur [18], and for the calculation of ploughing forces by Das and Dundur [19], Zou and Seethaler [20], and Waldorf et al. [21]. Predictive models for cutting forces using finite element analysis has been developed by Iwata et al. [22], Kim et al. [23], Strenkowski and Athavale [24], Strenkowski and Carrol [25], Shinozuka et al. [26], Childs et al. [27], Iqbal et al. [28], Afsharhanace et al. [29], and Li et al. [30].

The model which is widely used in the analysis of the mechanics of metal machining, however, is the simple, highly idealized shear plane model of chip formation developed by Merchant [31]. This model can be used to calculate the cutting forces and chip tool contact lengths using force and moment equilibrium and, as shown by Atkins [32], can quantitatively explain the vast majority of the unexplained experimental observations through the consideration of the specific work of surface formation.

Merchant's analysis is an upper bound since the cutting forces in this case are calculated using a kinematically admissible velocity field. But, unlike the classical upper bound approach, where the forces are determined from the upper bound on power, the cutting and thrust forces in this case are computed using force balance. This is because the classical upper bound approach cannot be used in a straight forward manner to analyze metal forming processes when friction at the contact faces obeys Coulomb's law.

In this paper, an attempt has been made to evaluate the cutting forces in orthogonal machining, with flat chip formation, from the upper bound on power. This method used the upper bound inequality formulated by Prager and Hodge [33] and Drucker et al. [34] and its modification by Collins [35] for the coulomb friction condition. In the following sections, Collin's "Generalised Upper Bound Technique" is briefly explained and its application for the construction of the hodograph in forming problems involving coulomb friction is elaborated. The equations for power consumption in the chip separation process were set up using the modified hodograph that was constructed following the above technique. The cutting force was calculated by equating the work of cutting with the above power. A procedure for the calculation of the tool's friction force from the energy consideration has also been presented. Theoretical cutting forces, thrust forces, tool/chip contact lengths, and chip thickness ratios were determined under different cutting conditions. These are also compared with some experimental results that are available in the literature [36]. It was seen that the results obtained using the present analysis are in agreement with those calculated using force balance [31].

## 2. The Generalized Upper Bound Technique

The upper bound theorem states that "among all the kinematically admissible strain rate fields, the actual one minimizes the expression [33,34]:

$$J^* = \frac{2}{\sqrt{3}}\sigma_0 \int_V \sqrt{1/2\,\varepsilon_{ij}\varepsilon_{ij}}\,dV + \frac{\sigma_0}{\sqrt{3}}\int_{S_F} |\Delta v|\,ds - \int_{S_T} T_i v_i ds''$$  (1)

In the above equation, the first term on the right represents the volumetric work, the second term stands for the power consumed at the surfaces of velocity discontinuity, and the third term provides the traction power.

For metal working operations undergoing deformation under the shear friction condition, ($\tau = mk$), the traction power due to interface friction can be easily calculated, and the upper bound load can be determined using Equation (1). However, under the Coulomb friction condition ($\tau = \mu p$), the determination of the friction work and, hence, the upper bound is not so simple as the friction stress in this case is a function of the local die pressure, which is unknown. For such problems, however, approximate estimates of the upper bound load may be obtained using several simplifying assumptions [37,38].

For a rigid-plastic material undergoing deformation under an externally applied load, Tu, the above upper bound energy dissipation equation may be stated as [35]:

$$\int T_u u^* dS_u \leq \int_V \sigma_{ij}^* \dot{\varepsilon}_{ij}^* dV + \frac{\sigma_o}{\sqrt{3}} \int_{S_F} |\Delta v| ds_F - \int_{S_T} T_i v_i dS_T \tag{2}$$

Collins [35] argued that the above inequality holds "irrespective of whether $u^*$ is incompatible with the rigid motion of the tool piece on $S_u$. In other words, $u^*$ does not have to be a kinematically admissible velocity field. Its only requirement is that, its component in the direction of Tu should be constant so that it can be taken outside the integral sign". This requires the chosen velocity field to be such that the velocity of the deforming material at the die/metal interface lies on the line normal to the direction of the externally applied surface traction $T_u$ on $S_u$. Collins applied this technique to compute the upper bound loads for plane strain compression and extrusion and found reasonable agreement with the corresponding slip-line field solutions [35].

### 3. Upper Bound Analysis of Orthogonal Machining

We considered the tool work system in the orthogonal machining of a work piece of width "b" with a flat, rigid, cutting tool of rake angle "$\alpha$", as shown in Figure 1a. Due to the chip flow along the tool face, the tool was subjected to the friction force "F" and the normal force "N", as indicated in Figure 1b. $N_s$ and $F_s$ represent the normal and shear forces on the shear plane, and $F_c$ and $F_{th}$ represent the tool cutting and thrust forces, respectively. For force-free chip streaming, the resultant of the above three pairs of forces must be equal. This is represented by the force R, shown in Figure 1b, and is in the Merchant's circle diagram, as indicated in Figure 1c. R is inclined horizontally at an angle ($\lambda$-$\alpha$), where $\lambda$ denotes the friction angle ($\mu = \tan\lambda = F/N$—$\mu$ is the coefficient of friction at the chip/tool interface).

Referring to Figure 1b, it could be seen that:

$$L_S = t_o / \mathrm{Sin}\phi \tag{3a}$$

$$F_C = R \cos(\lambda - \alpha) \tag{3b}$$

$$F_{th} = R \sin(\lambda - \alpha) \tag{3c}$$

where $L_S$ is the length of the shear plane, OA, to the uncut chip thickness, and $\phi$ is the shear plane angle.

Figure 1d represents the conventional hodograph for the problem where $v_c$ represents the cutting velocity, $v_s$ denotes the shear plane velocity, and $v_{chip}$ designates the chip velocity.

Considering Figure 1b,d, the upper bound energy dissipation equation for the problem could be written as:

$$F_c v_c = F_s . v_s + F . v_{chip} \tag{4}$$

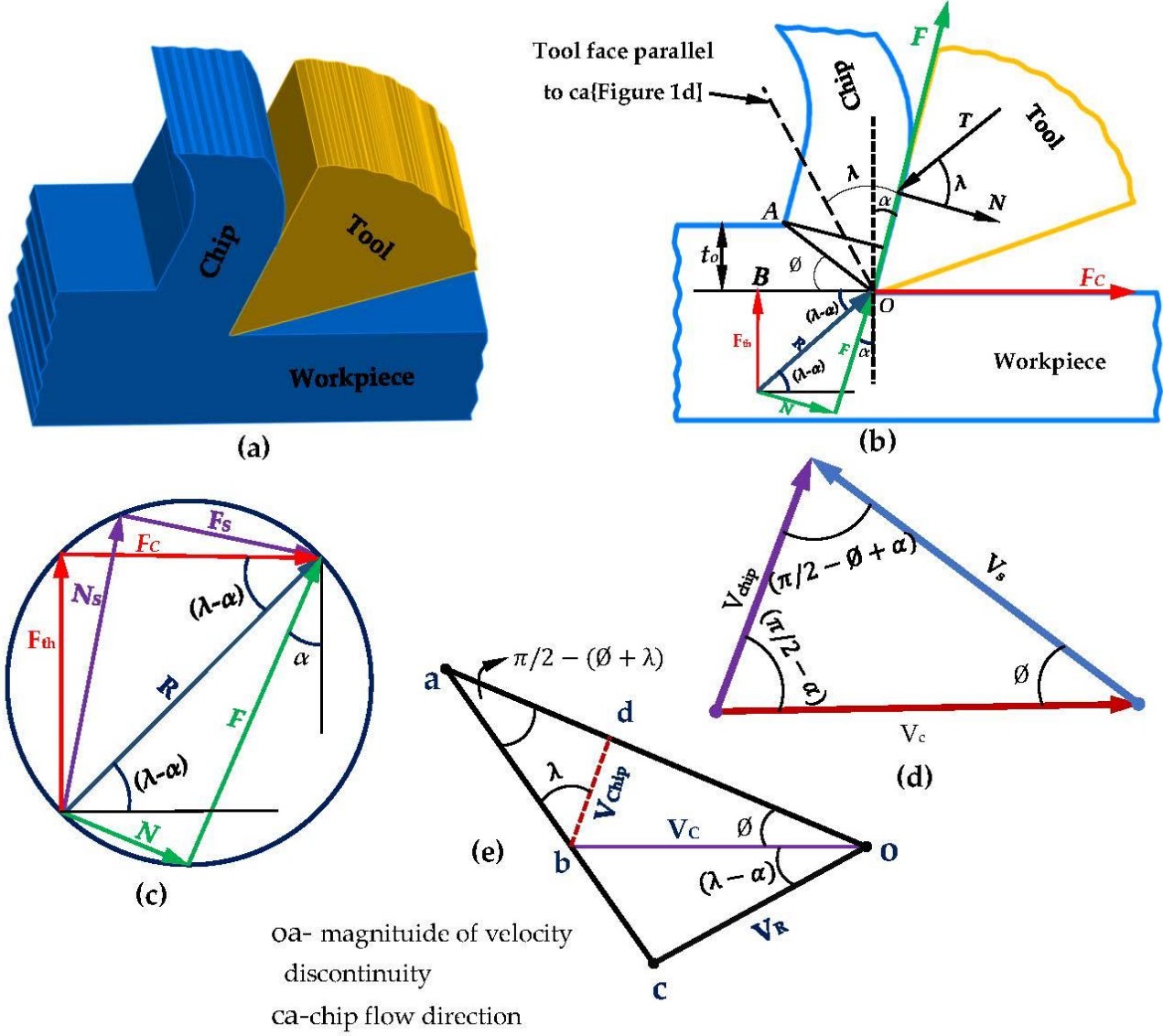

**Figure 1.** (**a**) Orthogonal machining—3D view. (**b**) Model of chip formation (obtained from the authors of [31]). (**c**) Force plane diagram. (**d**) Conventional hodograph. (**e**) Modified hodograph.

The shear force $F_s$ is given by:

$$F_s = k.L_s.b \tag{5a}$$

where '$k$' is the yield stress in the shear of the work material, and ($L_s.b$) is the area of the shear plane. Referring to Figure 1d, it may be seen that $v_c$, $v_s$, and $v_{chip}$ are related through the following equations:

$$v_s = v_c.\cos\alpha / \cos(\phi - \alpha) \tag{5b}$$

$$v_{chip} = v_c.\sin\phi / \cos(\phi - \alpha) \tag{5c}$$

However, $F$ in Equation (4) is unknown. Hence, this equation cannot be used to determine $F_c$.

Merchant [31] computed the upper bound on $F_c$ by considering the "force balance" (Figure 1c). This was possible as the surfaces of velocity discontinuity, such as the shear plane OA, and the tool face, in this case, were assumed to be straight. This implied that the volumetric work was zero (first term in the right of Equation (1)). However, when dealing with the problems where these surfaces are curved, and where the volumetric work is not

equal to zero, the method of "force balance" cannot be used for the calculation of the upper bound load. For these problems, one must refer to the generalized upper bound technique for the above analysis.

The modified hodograph for the problem is shown in Figure 1e. This has been drawn along the lines suggested by Collins [35]. In this figure, $oc = (v_R)$ represents the velocity in the direction of the force R. The line *oa*, which has been drawn parallel to the shear plane OA, meets the normal to the line *oc* at *a*. Therefore, this can be outlined as:

$$v_c = v_R / \cos(\lambda - \alpha) \tag{6a}$$

$$oa = v_R / \cos(\phi + \lambda - \alpha) \tag{6b}$$

where '*oa*' represents the magnitude of the velocity discontinuity suffered by the material on crossing the shear plane. The rate of internal energy dissipation is $kbL_s oa$, which should be equal to the work performed by the tool. Hence, this can be written as:

$$Rv_R = F_c.v_c = kL_s.boa \tag{7}$$

Substituting for $L_s$ from Equation (3a) and for oa from Equations (6b) and (7), this equation can be finally written as:

$$F_c = kt_o b \mathrm{Cos}(\lambda - \alpha)/(\mathrm{Sin}\phi.\mathrm{Cos}(\phi + \lambda - \alpha)) \tag{8}$$

The above equation was derived by Merchant in the following manner. Referring to Figure 1c, we could write the following:

$$F_c = R\mathrm{Cos}(\lambda - \alpha) \tag{9}$$

$$F_s = R\mathrm{Cos}(\phi + \lambda - \alpha) \tag{10}$$

Hence, these equations could ultimately be written as:

$$F_c = \frac{F_s \mathrm{Cos}(\lambda - \alpha)}{\mathrm{Cos}(|\phi + \lambda - \alpha)} \tag{11}$$

When substituting Equation (5a) into Equation (11), we arrived at Equation (8).

Thus, the relation for $F_c$ derived from the present method was the same as that derived by Merchant [31] from the "force balance".

Making $dF_c/d\phi = 0$, we arrived at the well-known shear angle relation:

$$2\phi + \lambda - \alpha = \pi/2 \tag{12}$$

It may be seen that the modified hodograph shown in Figure 1e is that for machining with a frictionless tool, inclined to the original tool face at an angle "λ" (Figure 1b).

The total energy during machining is partly dissipated due to shear along the shear plane and partly due to friction along the tool face. Therefore:

$$F_s.oa = F_s.v_s + F.v_{chip} \tag{13}$$

The line *bd* in Figure 1e is drawn parallel to the tool face. The triangle obd in Figure 1e is similar to that shown in Figure 1d. Thus, $od = v_s$ and $bd = v_{chip}$. Hence:

$$F.v_{chip} = F_s.(oa - od) = F_s.ad$$

Referring to triangle abd in Figure 1e, it may be seen that:

$$ad = bd \sin \lambda / \cos(\phi + \lambda - \alpha) \tag{14}$$

Hence, this indicates that: $F.v_{chip} = F.bd = F_s.bd. \sin \lambda / \cos(\phi + \lambda - \alpha)$ or

$$F = F_s. \sin \lambda / \cos(\phi + \lambda - \alpha) \tag{15}$$

In this manner, the friction force $F$ can be calculated from the hodographs 1(d) and 1(e) without referring to Merchant's force circle (Figure 1c).

## 4. Results and Discussion

The upper bound theorem, states that "for a plastically deforming medium, if a velocity field could be found, that is volume preserving, and is compatible with the velocity boundary conditions on the tool or die, then the force calculated from such a velocity field would be higher than the actual load" [4,33]. This load can be estimated by considering the energy dissipation rate for this velocity field (energy balance, Equation (1)).

For problems where the deformation energy cannot be estimated a priori, such as those involving coulomb friction, an alternative method has been used to compute the upper bound load. This method has been successfully used when the problem under consideration is one of plane strain, and which involves only planar boundaries, such as a flat tool face and straight surfaces of velocity discontinuity. This method considered the equilibrium of forces on elemental triangular elements to calculate the forming forces. This method was used by Merchant [31] to compute the cutting and thrust forces in orthogonal cutting, and by Westwood and Wallace [39], Green and Wallace [40], and Green et al. [41] for the upper bound analysis of tube drawing and hot and cold rolling.

In 1969, Collins [35] generalized the above upper bound formulation, such that it can be used to compute the deformation energy for metal forming problems involving Coulomb friction (Equation (2)).

In this paper, the problem of free-chip orthogonal machining was analyzed using the method proposed by Collins [35]. The cutting force for this case was calculated from the energy dissipation rate obtained from a velocity field that was constructed in the manner suggested above.

It must be mentioned here that the upper bound theorem was formulated in terms of energy. Collin's method follows this approach. It is more general, can be applied to both plane strain and axisymmetric problems, and can deal with both continuous [42] and discontinuous velocity fields. However, the method of 'force balance' has only limited application.

Theoretical results for cutting forces, thrust forces, chip/tool contact lengths, and chip thickness ratios as functions of the tool rake angle and the coulomb coefficient of friction, calculated using the present analysis, are presented in Figures 2–9, where these are compared with some experimental values that have been reported in the literature [43]. The non-dimensional cutting and thrust forces from the experimental data have been computed using the following equations:

$$\frac{F_c}{kt_0 b} = \frac{\cos(\lambda - \alpha)}{\sin \phi \cos(\phi + \lambda - \alpha)} \tag{16}$$

$$\frac{F_{th}}{kt_0 b} = \frac{\sin(\lambda - \alpha)}{\sin \phi \cos(\phi + \lambda - \alpha)} \tag{17}$$

$$\mu = \tan \lambda = \frac{F_{th} + F_c \tan \alpha}{F_c - F_{th} \tan \alpha}$$

$$\cot \phi = \frac{\zeta - \sin \alpha}{\cos \alpha} \tag{18}$$

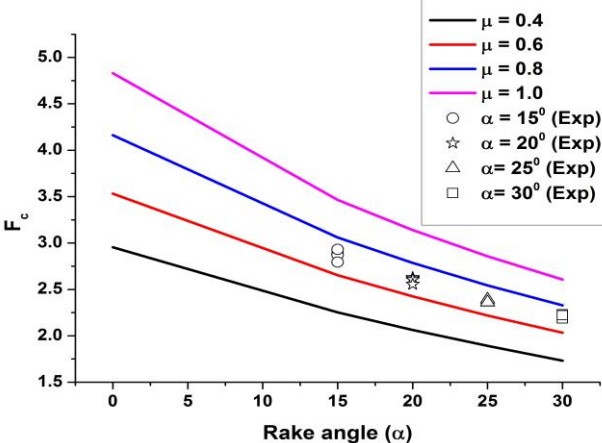

**Figure 2.** Comparison between the theoretically calculated cutting forces and the experimental values [43]. $v_c$ = 90.8 ft/min (27.68 m/min).

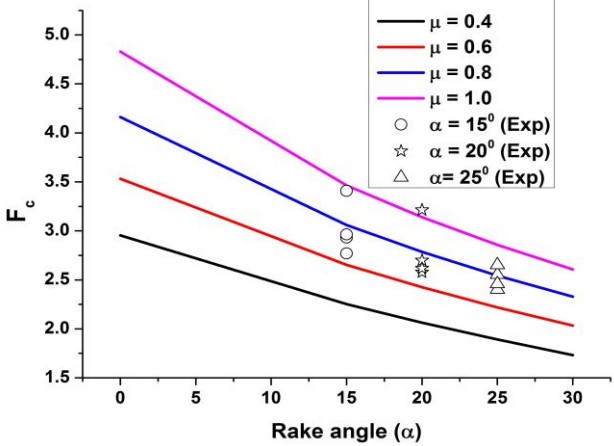

**Figure 3.** Comparison between the theoretically calculated cutting forces and the experimental values [43]. $v_c$ = 170.8 ft/min (52.06 m/min).

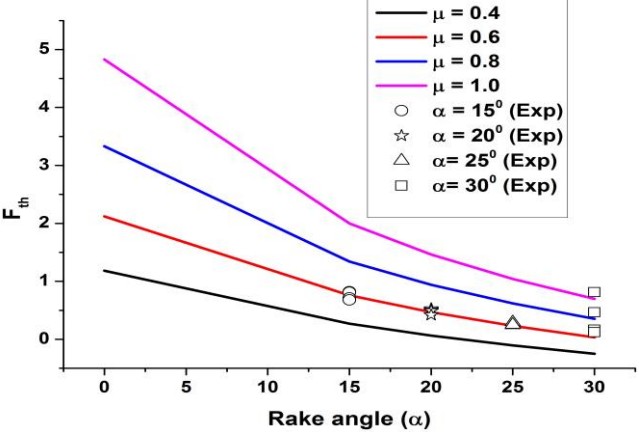

**Figure 4.** Comparison between the theoretically calculated thrust forces and the experimental values [43]. $v_c$ = 90.8 ft/min (27.68 m/min).

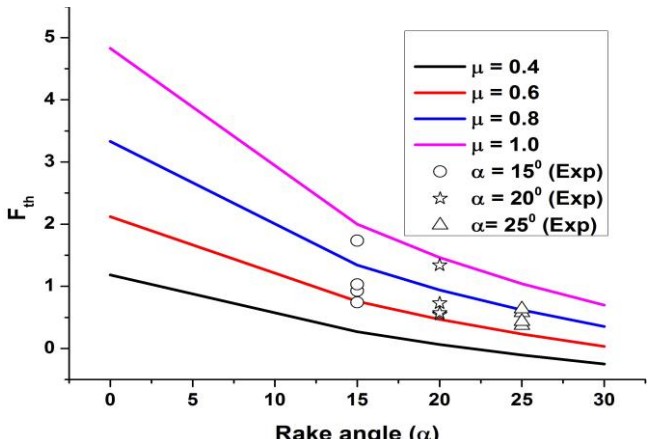

**Figure 5.** Comparison between the theoretically calculated thrust forces and the experimental values [43]. $v_c$ = 170.8 ft/min (52.06 m/min).

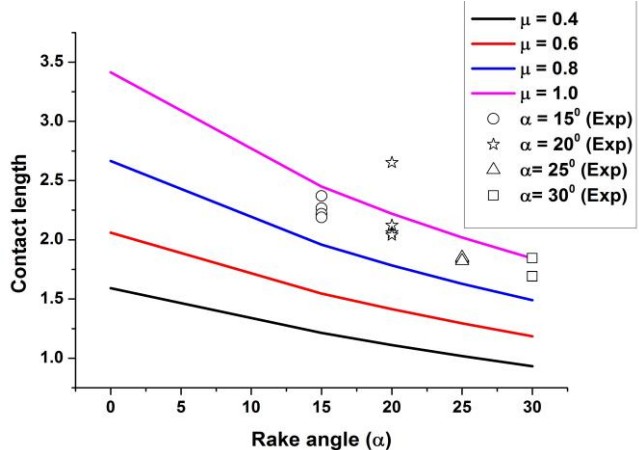

**Figure 6.** Comparison between the theoretically calculated tool/chip contact lengths and the experimental values [43]. $v_c$ = 90.8 ft/min (27.68 m/min).

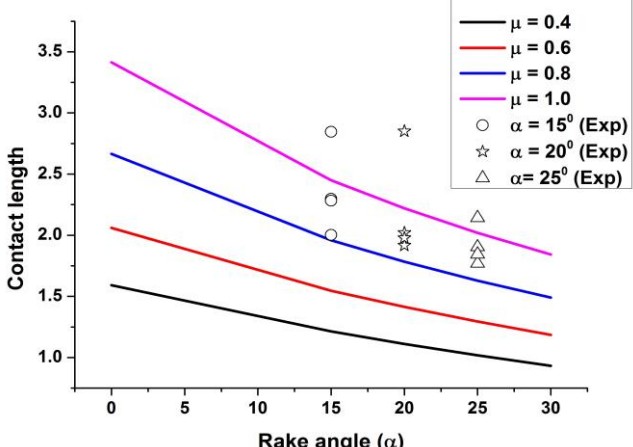

**Figure 7.** Comparison between the theoretically calculated tool/chip contact lengths and the experimental values [43]. $v_c$ = 170.8 ft/min (52.06 m/min).

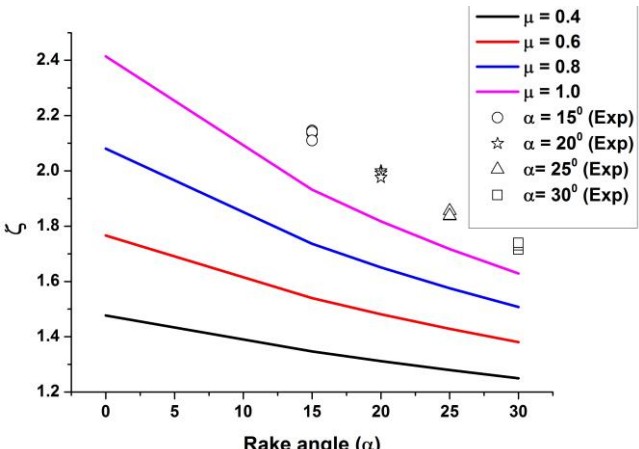

**Figure 8.** Comparison between the theoretically calculated chip thickness ratios and the experimental values [43]. $v_c$ = 90.8 ft/min (27.68 m/min).

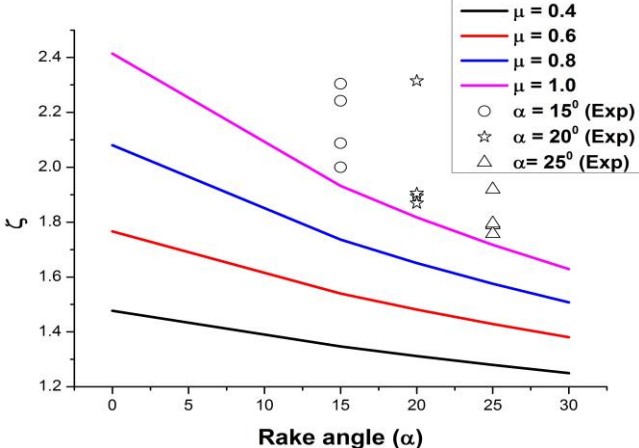

**Figure 9.** Comparison between the theoretically calculated chip thickness ratios and the experimental values [43]. $v_c$ = 170.8 ft/min (52.06 m/min).

The tool/chip contact length and chip thickness ratio can be calculated using Equation (12). These may be written as:

$$\text{Tool/chip contact length} = \frac{\cos\phi}{\cos\lambda\sin\phi} \tag{19}$$

$$\text{Chip thickness ratio} = \frac{\cos(\phi - \alpha)}{\sin\phi} \tag{20}$$

In deriving the theoretical results presented in Figures 2–9, the following two factors have not been taken into account:

(i)   The over-stressing limits and the limits of validity, as suggested by Hill [43];
(ii)  The formation of built-up edges take place at high friction conditions [2].

Together, the above figures indicate that the present calculated values do not compare favorably with the experimental values. Of note, the theoretical Coulomb coefficient of friction was found to be much higher than those found experimentally.

The reason for this may be that the nature of friction at the tool/chip interface is one of adhesion and not coulomb, as is normally assumed. The variation of the normal and shear stresses, as obtained from a split-tool apparatus, is shown in Figure 10 [44]. This figure indicates that neither the normal nor the shear stresses are uniformly distributed on the

tool/chip contact length. Hence, the calculation of μ from Equation (18) may not be correct. However, even under the adhesion friction condition, there may still be a considerable scatter of experimental results, as outlined in the study published by the authors of [6]. Such discrepancy between the theoretical and the experimental results has also been reported earlier by other authors [4,5]. This demonstrates the non-unique nature of the machining process (Hill [43]).

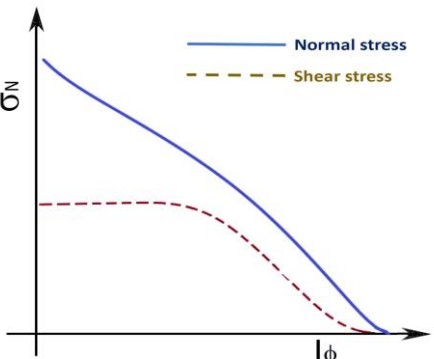

**Figure 10.** Variation of shear and normal stress at the tool/chip interface [44].

Figures 6–9 indicate that the experimental values of the chip thickness ratios and the tool/chip contact lengths were higher than those predicted theoretically with the present analysis. This may be due to the fact the shear angle relation assumed in the present analysis ($2\phi + \lambda - \alpha = \pi/2$) was only approximate. As a matter of fact, the variation of $\phi$ with (λ-α) shows a considerable level of scatter, as outlined in the studies published by the authors of [4,42].

## 5. Conclusions

This paper presents an alternative method of analysis of free-chip orthogonal machining, where the cutting forces are calculated from the energy dissipated during the chip separation process. The method is based on the classical upper bound formulation and its modification by Collins to deal with the problems involving Coulomb friction. The cutting forces, thrust forces, the tool/chip contact lengths, and the chip thickness ratios are calculated for different rake angles and friction conditions. The theoretical values are also compared with several experimental results that are available in the published literature. The agreement between the two, however, was not found to be satisfactory. This may be due to the non-unique nature of the machining process.

**Author Contributions:** Conceptualization, N.S.D. and H.J.; methodology, N.S.D.; software, H.J. and B.H.; validation, H.J., B.H., N.A.A. and S.A.; formal analysis, N.S.D.; writing—original draft preparation, H.J., B.H. and S.A.; writing—review and editing, N.S.D., H.J., B.H., K.M., N.A.A. and S.A.; supervision, N.S.D. and B.H.; project administration, B.H. and S.A.; funding acquisition, B.H., S.A. and N.A.A. All authors have read and agreed to the published version of the manuscript.

**Funding:** This work was supported and funded by the Deanship of Scientific Research at Imam Mohammad Ibn Saud Islamic University (IMSIU) (grant number IMSIU-RG23023).

**Data Availability Statement:** Not applicable.

**Conflicts of Interest:** The authors declare no conflict of interest.

## Nomenclature

b = width of the work piece
J* = total energy dissipation rate
k = yield stress in shear of the work material
$t_o$ = uncut chip thickness
$t_{chip}$ = chip thickness
m = friction factor for the shear friction condition
u* = prescribed velocity on the surface $S_u$
$v_c$ = cutting velocity
$V_{chip}$ = chip velocity
$v_i$ = velocity in the direction of the specified traction Ti on the surface $S_T$
$v_R$ = velocity in the direction of the resultant tool force R
$|\Delta v|$ = velocity jump across a surface of velocity discontinuity
F, N = friction force and normal force on the tool face (Figure 1a,b)
$F_c$, $F_{th}$ = tool cutting force and the thrust force (Figure 1a,b)
$F_s$, $N_s$ = shear force and normal force on the shear plane (Figure 1a,b)
Ls = length of the shear plane
Lc = tool/chip contact length
R = resultant of the above three pairs of forces (Figure 1a,b)
$S_F$ = surface across which there is a discontinuity in velocity (Equation (1))
$S_T$ = surface where traction Ti is specified (Equation (1))
$S_u$ = surface with specified velocity u* (Equation (1))
$T_i$ = surface traction on the surface $S_T$ (Equation (1))
$T_u$ = traction on surface Su (to be calculated) (Equation (1))
$\alpha$ = tool rake angle
$\varepsilon_{ij}$ = strain rate
$\lambda$ = angle of friction (tan$\lambda$ = $\mu$)
$\mu$ = coefficient of friction at chip-tool interface
$\phi$ = angle made by the shear plane with the direction of the tool travel
$\sigma_o$ = yield stress in compression of the work material
$\xi$ = chip thickness ratio
$\tau$ = frictional traction

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
