# Peer review of "An Upper Bound Energy Formulation of Free-Chip Machining with Flat Chips and an Alternative Method of Determination of Cutting Forces without Using the Merchant’s Circle Diagram"

_machines, doi:10.3390/machines11090853_

Round 1
Reviewer 1 Report
The manuscript had described an upper bound analysis of free chip machining, the tool cutting and friction forces had been determined from the deformation energy dissipated during the chip separation process. The work is very important for calculating the cutting force, thrust force and contact length for different cutting conditions. The manuscript could be published after the minor modified:
(1) There are 9 keywords, they are too much, 5 keywords is well.
(2) The sign of “φ” should be the sign of “Ø” in equation (3a), (6b), (8), (9), line 263. The whole paper should be rechecked to avoid the mistake like this.
(3) The description of “Contact Length” should be rechecked in equation (16).
(4) Is there any way to calculated the adhesion friction value in Line 251? The theoretically calculated value in Fig.6 to Fig.9 would shown less discrepancy, but this is just a suggestion.
(5) There is no explaination or description of Fig. 10 in section 4.
(6) Section 5 should be the conclusion of the innovative work of this manuscript, it should be concise and to the point. The references [30] and [37] shouldn’t be shown in “Conclusions”.
(7) The published date of most references is too old, there are only 5 papers is published after the year of 2010.
The quality of English language is well, just minor editing of English language is required.
Reviewer 2 Report
1. The "The results calculated from the present method of” energy balance” are also found to agree with those obtained by Merchant using the principle of ”force balance”" in the abstract does not see validation in the manuscript and needs to be validated by you for comparison.
2. The legends of figures 3,4,5,8,9 are missing units.
3. Figure 2-9 Why the calculated conditions do not match the experimental conditions, e.g., Why experiments and theoretical calculations of chip thickness do not agree.
4. Suggested citations for the following papers:
Wang H , To S , Chan C Y ,et al. A study of regularly spaced shear bands and morphology of serrated chip formation in microcutting process[J].Scripta Materialia, 2010, 63( 2):227-230.DOI:10.1016/j.scriptamat.2010.03.059.
Jing C , Wang J , Zhang C ,et al. Influence of size effect on the dynamic mechanical properties of OFHC copper at micro-/meso-scales[J].The International Journal of Advanced Manufacturing Technology, 2022(7/8):120.
Zhang J , Lee Y J , Wang H .Microstructure evaluation of shear bands of microcutting chips in AA6061 alloy under the mechanochemical effect[J].Journal of Materials Science & Technology, 2021.
1. It is recommended that you need check your grammar and spelling.
Reviewer 3 Report
1. Page 2 lines 44 - 56 - literature analysis must include a description of what exactly each literature item represents, what research has been done and what results the authors of the research have obtained.
2. In my opinion, there is a need for experimental verification of the research results. Because, as the authors themselves write, comparison with some experimental values is unsatisfactory.
3. Conclusions are very general. No reference to the results obtained. There is no information about the exact discrepancy between the obtained results and the results obtained by other scientists.
4. The article lacks descriptions of Figures 2 - 9.
5. Please change vc unit to m/min
6. Please unify the legends in figures 2 - 9
Author Response
lease see the attached doc file

Reviewer 4 Report
Numbers means a number of the verse:
16. should be "Coulomb friction". Please correct in the whole paper.
25. should be "free-chip machining"
88. number of the formula should be aligned to the right. Please correct in the whole paper.
93. The formula is pasted in a bad way.
Fig.1a Is not should be "orthogonal cutting"?
Why did the Authors use the old bibliography instead of the new one? In many newest papers authors research additionally e.g. chip shapes and the influence of machining technological parameters on the state of the surface layer (e.g. papers from only one team who research similar phenomena:
Investigations on the chip shape and its upsetting and coverage ratios in partial symmetric face milling process of aluminium alloy AW-7075 and the simulation of the process with the use of FEM, Chodor J., Zurawski L., ADVANCES IN MECHANICS: THEORETICAL, COMPUTATIONAL AND INTERDISCIPLINARY ISSUES, Page 121-124, Published 2016
Author Response
lease see the attached file

Round 2
Reviewer 2 Report
None
Reviewer 3 Report
Thank you for considering my comments